# Interleukin 6 and Aneurysmal Subarachnoid Hemorrhage. A Narrative Review

**DOI:** 10.3390/ijms22084133

**Published:** 2021-04-16

**Authors:** Matthias Simon, Alexander Grote

**Affiliations:** Department of Neurosurgery, Bethel Clinic, University Hospital, University of Bielefeld Medical School, 33617 Bielefeld, Germany; matthias.simon@evkb.de

**Keywords:** subarachnoid hemorrhage, cerebral vasospasm, DCI, neuroinflammation, IL-6

## Abstract

Interleukin 6 (IL-6) is a prominent proinflammatory cytokine. Neuroinflammation in general, and IL-6 signaling in particular, appear to play a major role in the pathobiology and pathophysiology of aneurysm formation and aneurysmal subarachnoid hemorrhage (SAH). Most importantly, elevated IL-6 CSF (rather than serum) levels appear to correlate with delayed cerebral ischemia (DCI, “vasospasm”) and secondary (“vasospastic”) infarctions. IL-6 CSF levels may also reflect other forms of injury to the brain following SAH, i.e., early brain damage and septic complications of SAH and aneurysm treatment. This would explain why many researchers have found an association between IL-6 levels and patient outcomes. These findings clearly suggest CSF IL-6 as a candidate biomarker in SAH patients. However, at this point, discrepant findings in variable study settings, as well as timing and other issues, e.g., defining proper clinical endpoints (i.e., secondary clinical deterioration vs. angiographic vasospasm vs. secondary vasospastic infarct) do not allow for its routine use. It is also tempting to speculate about potential therapeutic measures targeting elevated IL-6 CSF levels and neuroinflammation in SAH patients. Corticosteroids and anti-platelet drugs are indeed used in many SAH cases (not necessarily with the intention to interfere with detrimental inflammatory signaling), however, no convincing benefit has been demonstrated yet. The lack of a robust clinical perspective against the background of a relatively large body of data linking IL-6 and neuroinflammation with the pathophysiology of SAH is somewhat disappointing. One underlying reason might be that most relevant studies only report correlative data. The specific molecular pathways behind elevated IL-6 levels in SAH patients and their various interactions still remain to be delineated. We are optimistic that future research in this field will result in a better understanding of the role of neuroinflammation in the pathophysiology of SAH, which in turn, will translate into the identification of suitable biomarkers and even potential therapeutic targets.

## 1. Introduction

Subarachnoid hemorrhage (SAH) is a major cause of cerebrovascular morbidity and mortality in often young patients. De Rooij et al. reported an overall incidence of SAH of approximately 9 per 100,000 person years, with a large regional variability (e.g., 22.7 and 19.7 per 100,000 in Japan and Finland vs. 4.2 per 100,000 in South America) [1]. Roughly 85% of cases are caused by a ruptured cerebral aneurysm [2]. Aneurysm occlusion and management of the sequelae of the hemorrhage account for a major part of the vascular neurosurgery and neuroradiology workload.

One of the major complications following SAH is the occurrence of a peculiar clinical syndrome consisting of secondary neurological impairment that often reflects the involvement of major arterial vascular territories, which is quite often responsive to induced arterial hypertension. This is usually accompanied by major artery narrowing, which is diagnosed by conventional, CT or MR angiography, or transcranial Doppler (TCD) ultrasound studies. Functional CT and MR imaging often reveals perfusion deficits, which in many cases result in manifest cerebral infarctions. This condition has historically been termed “vasospasm”, reflecting the mechanistic concept of arterial narrowing, resulting in perfusion deficits and ischemia, and ultimately in infarctions. However, the association between angiographic and clinical findings is far from perfect, and a more modern view reserves the term vasospasm for angiographic findings and the clinical condition is termed delayed cerebral ischemia (DCI) or delayed ischemic neurological deficit (DIND) [2,3,4,5,6].

The pathophysiology of aneurysm growth and rupture, SAH and in particular of DCI [2,4] is likely complex and involves genetic factors [7,8,9], formation of microthrombi [10,11], cortical spreading depolarizations [12], as well as small, medium and large artery spasms, and—importantly—(neuro)inflammation [13]. IL-6 is a prominent cytokine and a major neuroinflammatory mediator [14]. Neuroinflammation following SAH has attracted considerable attention from researchers, and many groups have focused specifically on the role of IL-6 signaling. This review attempts to summarize our current understanding of how IL-6 contributes to the various aspects of SAH pathophysiology. We will also try to present an overview of potential clinical applications. There indeed seems to be a considerable body of data to suggest IL-6 as a potential biomarker and even molecular target for anti-neuroinflammatory therapeutic interventions. However, many issues remain unresolved despite many years of research.

## 2. IL-6 and Neuroinflammation

Neuroinflammation refers to a variety of inflammatory responses to noxious stimuli targeting the brain and/or spinal cord. The resident microglia play a central role in all neuroinflammatory processes by producing inflammatory mediators such as chemokines, cytokines and reactive oxygen species [15]. Defining neuroinflammation is difficult and the use of a somewhat loose concept of neuroinflammation as the presence of inflammatory mediators in the CNS together with microglia activation has resulted in grouping detrimental pathologies together on the one hand with physiological regulatory processes on the other [15].

Neuroinflammation in general and IL-6 in particular are thought to play a very significant role in the pathophysiology of diverse disorders such as traumatic brain and spinal cord injury, neurodegenerative diseases such as Alzheimer’s and Parkinson’s syndrome, CNS infection and stroke [14,15]. Mathiesen and coworkers reported elevated CSF IL-6 levels in SAH patients in 1993 [16]. Importantly, only low levels of IL-6 are seen in the brain under non-pathological conditions, while a variety of diseases have been shown to be accompanied by high levels of IL-6 expression [14].

On the other hand, IL-6 signaling is involved in the physiological homeostasis of the brain (“euflammation”) [15]. The regulation of sleep, the stress response, pain, memory, aging and various repair processes [14,17] have all been shown to involve neuroinflammatory signaling and often also IL-6. Many investigations have documented that IL-6 signaling has protective and not only detrimental effects, e.g., IL-6 counteracts NMDA receptor-mediated excitotoxicity following brain ischemia and also promotes nerve regeneration. It is possible that the involvement of IL-6 signaling in both beneficial and physiological as well as pathological processes in the brain primarily reflects the activity of two different (“classical” and “trans”) signaling pathways [14].

## 3. IL-6 and SAH

### 3.1. Aneurysm Formation and Rupture

Neuroinflammatory signaling pathways may contribute to the formation, growth and eventual rupture of cerebral aneurysms. Lymphocyte and macrophage infiltration is a histopathological hallmark of intracerebral aneurysms, and many cytokines including IL-6, adhesion molecules, immunoglobulins, and complement factors have been detected in aneurysm tissues [18]. Shimada and coworkers showed that use of the peroxisome proliferator-activated receptor-γ (PPARγ) pioglitazone in a mouse model protects against aneurysm rupture, and that this was associated with a reduction of IL-6 mRNA levels [19]. Wajima et al. observed IL-6 expression in aneurysm tissues but not in control arteries, and IL-6 was found to promote rupture of experimental aneurysms in a murine estrogen-deficient model but not in wildtype animals [20]. Similarly, Chen et al. reported a role for MiR-21, the JNK signaling pathway and IL-6 in the formation and rupture of aneurysms in a mouse model [21]. An anti-inflammatory DPP-4 (dipeptidyl peptidase-4) inhibitor (anagliptin) has been shown to reduce aneurysm growth through NF-κ-B and ERK5 signaling involving IL-6 [22,23].

Contradictory findings have also been published. A recent study found differential (lower) expression of IL-6 and upregulation of other inflammatory mediators in cerebral aneurysm as compared to middle temporal artery tissues [24]. Sawyer and coworkers [25] observed much higher IL-6 tissue levels but lower rates of aneurysm formation and rupture in lymphocyte-depleted vs. wildtype mice. Hosaka et al. recently demonstrated that IL-6 and osteopontin are key downstream mediators of MCP-1-mediated aneurysm healing in an experimental murine model [26].

Finally, genetic polymorphisms of the *IL-6* gene have been associated with aneurysm formation in British, Chinese and other populations [27,28,29]. However, these findings have not been replicated in large GWA studies [7,8,9].

Together, these and other published data support the concept of IL-6 as a prominent player in the pathophysiology of cerebral aneurysms. The precise molecular pathways involved are unclear at this point, but at present several major signaling pathways have been implicated. In accordance with the observation of pleiotropic effects of IL-6 and neuroinflammation in the brain in general, it is likely that IL-6 may mediate both protective effects and healing as well as pathological and destructive signals in aneurysm pathobiology.

### 3.2. Early Brain Injury

Early brain injury describes the acute to subacute pathophysiology following an acute injury to the CNS within 72 h. While this concept was originally developed and validated in the context of traumatic brain injury research, it can and has been readily adapted to the study of SAH. Savarraj and coworkers were able to correlate elevated serum levels of IL-6 and various other inflammatory mediators with measures of the early brain injury parameters, i.e., brain edema scores based on imaging (CT) studies and the respective patient’s Hunt and Hess grade [30]. Other investigators have also described correlations or at least borderline associations between early serum and CSF IL-6 levels and the Hunt and Hess grade [31,32]. In our recent study, we also found somewhat higher IL-6 CSF levels in cases with Fisher grade 4 vs. 3 bleedings and in WFNS (World Federation of Neurological Societies) grade 5 SAH patients [33]. In summary, these data may point to a role for IL-6 signaling in the early inflammatory response to a SAH, i.e., in the pathophysiology of early post-SAH brain injury. Of note, IL-6 levels after traumatic brain injury are usually considerably lower than those after SAH [34]. It is possible that this reflects pathophysiological differences between early brain injury following trauma vs. SAH.

Since early brain injury has a heavy impact on the patients’ eventual functional prognosis, this may explain in part why early IL-6 levels have been associated with patient outcomes by some authors [32]. However, the most pertinent studies (Table 1) report much higher IL-6 levels >72 h, suggesting that IL-6 is also or even primarily involved in the later stages of the disease.

### 3.3. Delayed Cerebral Ischemia

A number of studies have investigated a potential correlation between elevated IL-6 levels and later DCI. In 1998, Gaetani et al. reported elevated levels of various cytokines including IL-6 in the cisternal CSF of aneurysm patients as well as a potential correlation with vasospasm [35]. Ever since, many other groups have also investigated IL-6 as a biomarker for DCI (Table 1).

Most investigators have published positive findings associating DCI with high CSF IL-6 levels [5,33,35,38,39,43,45,47,51,52,53,55] but also correlations with serum IL-6 levels [42,45,46,50,52,53,56]. IL-6 levels were found to usually peak in the first week following the hemorrhage. Investigations of both CSF and serum levels reported either concordant or positive finding only for the CSF analyses (Table 1). This may well reflect the much higher numerical values seen in the CSF when compared to serum. The latter finding is actually a very strong argument for the role of neuroinflammation vs. that of a systemic inflammatory response in SAH pathophysiology. IL-6 levels in the extracellular fluid of the brain can be measured using micro-dialysis. Sarrafzadeh et al. reported concordant and positive findings for micro-dialysis and CSF (but not serum) studies associating high IL-6 levels with later DCI [41]. We recently had the opportunity to study IL-6 CSF and serum levels in an unselected cohort of 82 SAH patients [33]. There were very large inter-individual variations. However, we observed a characteristic time course of daily IL-6 CSF levels with a peak between days 4 and 14 including a maximum on day 5 after SAH that very much resembled the risk vs. time curve for DCI. DCI is commonly diagnosed between days 4 and 14 following SAH, and only very rarely in the third week after the bleed. Individual CSF peak levels were found to correlate significantly with the occurrence of DCI.

There are some notable discrepancies between the various published studies. Differential findings may result from different study designs and timing of IL-6 level determinations. Since CSF IL-6 level measurements are not part of the clinical routine in most centers, investigations of CSF findings usually require prospective patient enrollment and (variably) predefined timepoints for IL-6 testing. Some publications report non-consecutive cases and patient selection criteria are not always detailed, however, quite a few datasets come from prospective observational consecutive case series. Some studies use admission findings only, while others rely on serial measurements or even daily testing. Details can be found in Table 1. Different results in different patient cohorts can also be explained by assuming that IL-6 signaling is not causally or directly linked to DCI, and that increased IL-6 CSF levels following SAH are only a variable epiphenomenon reflecting the activity of other more relevant neuroinflammatory pathways.

Importantly, contradictory findings may also reflect problems with the proper definition of the endpoints of the study. A modern view of DCI will distinguish between angiographic vasospasm, the clinical DCI syndrome and secondary (DCI related or vasospastic) infarction [2,3,4,5,6,62,63]. We were able to investigate this issue in our series in more detail [33]. Interestingly, we found that the presumed association between IL-6 CSF level elevations and DCI in our series primarily reflected high IL-6 levels in cases that later developed DCI-related infarctions rather than an association between high IL-6 levels and the clinical syndrome (i.e., secondary neurological deterioration not otherwise explained) or angiographic vasospasm. We believe that this is a novel and potentially important finding, since these data suggest that IL-6 CSF levels can be used to identify patients with a high risk of infarcts, who therefore require aggressive treatment. It is also possible that in part, the clinical DCI syndrome and secondary vasospastic infarctions may indeed constitute different pathologies. These findings of course need confirmation by others.

In summary, there already seems to be a surprisingly large body of data that describes elevated CSF (and serum) IL-6 levels in at least a significant number of patients, who later on develop DCI and DCI-related infarctions. The pathophysiology behind this finding remains unclear. Nevertheless, these data underline the role of neuroinflammation in the pathophysiology of DCI. Attributing a central role to neuroinflammation in the pathogenesis of DCI will likely lead to very different paradigms of DCI prevention and treatment than the current emphasis on imaging-based diagnosis and angiographic treatment (i.e., intra-arterial spasmolysis).

### 3.4. EVD Related Infections

IL-6 signaling play a role in aseptic but also in septic inflammatory conditions. Indeed, IL-6 serum levels are used in many centers for the diagnosis and monitoring of septic conditions and this topic has even been subjected to a Cochrane review (albeit with inconclusive results) [64]. IL-6 is also a candidate biomarker for meningitis [65], however, its use in neurosurgical patients has been debated [66,67].

Many SAH patients suffer from hydrocephalus and need ventricular drainage, often for prolonged periods of time. Lenski et al. recently analyzed their experience of determining IL-6 CSF level in SAH patients with ventricular drains, which focused on both DCI and bacterial meningitis, and reported data that also suggested IL-6 as a potentially useful parameter for the diagnosis of CNS infections [55]. Other investigators have reported similar findings [67]. Our own investigation provided little supportive data [33]. These discrepancies may in part reflect the different timing of IL-6 level determinations in the various studies. Of note, in contrast to us and others, Lenski and coworkers were able to study daily IL-6 level determinations.

Different findings by different authors also highlight another central problem with the use of IL-6 as a marker for bacterial infections in the context of acute SAH besides timing, i.e., the prominent role of IL-6 signaling in both bacterial and aseptic inflammatory conditions, which often coexist. In patients with acute SAH, the heavy impact of DCI on IL-6 levels as outlined above may well prove to be too dominant to allow for the simple routine use of IL-6 as a marker of bacterial CNS infections.

### 3.5. Clinical Outcomes

Many studies describing IL-6 levels in SAH patients have also attempted to correlate IL-6 levels with other clinical features of SAH. A possible correlation between IL-6 levels and clinical outcome as described by many groups including ourselves [33], might be of particular importance [31,32,40,46,47,48,49,53,59]. DCI, and more specifically, DCI-related infarctions have a major influence on patient outcome [2,3,6,62,68]. This may explain much of the association between outcomes and IL-6 levels. As already mentioned above, early brain injury also impacts heavily on patient outcomes and also results in elevation of the IL-6 level.

Some studies found IL-6 levels predictive for later shunt-dependency [53,69,70]. We and others have not been able to replicate this finding [33,71]. Interestingly, a recent literature review concluded that IL-6 levels are often elevated in the CSF of patients with posttraumatic and other forms of hydrocephalus [72].

## 4. Perspectives

### 4.1. IL-6 as a Biomarker in SAH Patients?

Identification of a diagnostic biomarker would be major step forward in the management of DCI. The clinical syndrome of DCI can only be diagnosed in patients who are not intubated and sedated, and the diagnosis of DCI in such cases rests on transcranial Doppler sonography, DSA, CTA and MR studies [2,3,4,5,6,62,63,73,74]. However, the correlation between clinical symptoms, eventual infarctions and angiographic and imaging findings is far from perfect. A significant number of sedated SAH patients are possibly overtreated based on adjunct studies such as TCD and CTA, and in others, DCI is only diagnosed after an infarction has already been incurred. Hence, the association between increased CSF IL-6 levels and DCI reported by many investigators (Table 1) may be a good starting point for the development of a novel biomarker. Of note, the evidence that serum IL-6 determinations are useful is less robust, however, access to CSF is only easily possible in a proportion of SAH patients, i.e., cases with ventricular drains. Also, serial CSF sampling may carry a relevant risk of iatrogenic meningitis [75]. In selected cases, undergoing multimodal monitoring including placement of micro-dialysis catheters and IL-6 testing of extracellular brain fluid can be considered. However, the available data do not suggest that this approach is superior to more conventional CSF analyses [34,41].

Our own findings associating high IL-6 levels with secondary (DCI related) infarction rather than the clinical DCI syndrome may have important implications for the potential role of IL-6 as a biomarker [33]. The occurrence of imaging-proven secondary infarctions is a clinically very relevant and robust criterion for DCI; however, it can only be used in a post hoc setting [3,6,63]. If elevated IL-6 levels are indeed proven to identify cases with a high risk of infarction, early IL-6 determinations may help to select cases for particularly aggressive or possibly even differential DCI management. As mentioned above, the association between elevated IL-6 levels and secondary infarction, but not vasospasm or the clinical DCI syndrome, may point to pathophysiological differences between the respective patient groups. Different pathophysiologies will eventually require differential treatment paradigms.

The issue outcome prediction in the management of SAH patients probably also deserves more attention [31,32,33,40,46,48,49,50,53,59]. The treating neurosurgeon is commonly confronted with decision-making in clinical settings in which the patient’s functional prognosis plays an important role. Clinical parameters and imaging findings alone often do not suffice to decide if it is appropriate to apply aggressive treatment for a complication incurred during the intensive care unit (ICU) stay or through aneurysm, DCI or hydrocephalus management. As outlined above, some data suggest a role for IL-6 as an outcome predictor, possibly summing up all sorts of relevant brain injury following SAH. No patient in our series with a CSF IL-6 day 4–14 peak >35,000 pg/mL was found to have a mRS 0–2 (good) outcome [33]. Hostettler and coworkers applied decision tree analysis to a series of 548 SAH cases randomly divided into a derivation and a validation cohort in order to predict survival, functional outcomes, and shunt dependency. The day 1 serum IL-6 level was found to be an important differentiating factor for survival (but not functional outcome or delayed cerebral infarction prediction). Other inflammatory markers such as procalcitonin (PCT and CRP) also figured prominently as differentiating factors [76].

Any discussion of a potential role for IL-6 as a biomarker in SAH patients has to acknowledge the many limitations of the published data. All studies more or less aim to describe correlations between IL-6 levels (and other neuroinflammatory markers) and various clinical aspects of SAH rather than testing a prespecified hypothesis, that is, IL-6 levels on a predefined day predict DCI. The actual IL-6 levels reported vary between studies. Lack of specificity might also be an issue. Chaudhry et al. found an association between serum IL-6 levels not only with DCI and outcome, but also with a variety of CNS and systemic complications following SAH [53]. Patient selection bias has to be taken into consideration. The two largest series reported consist of only 138 and 149 cases, respectively (Table 1) [45,46].

As already mentioned above, the timing of IL-6 determination is also a crucial issue that must be resolved. In our own study we found a less than favorable balance between sensitivity and specificity of IL-6 testing for DCI prediction. Receiver operating characteristics (ROC) analysis and use of the Youden index showed that CSF IL-6 levels >44,500 pg/mL predicted DCI-related infarcts with a 94–98% specificity but only 46% sensitivity [33]. Lenski et al. reported much better results (sensitivity: 88%, specificity: 92%) when analyzing CSF IL-6 levels for DCI prediction after exclusion of cases with CSF infections. Of note, these authors tested a point-of-care paradigm [55].

In conclusion, it is probably fair to state that as of right now, IL-6 is at best, a candidate biomarker in SAH patients. Given the fact that the association between, e.g., DCI and high IL-6 levels has been investigated for quite some time without resulting in widespread IL-6 testing as part of clinical routines, it is possible that other markers of neuroinflammation might turn out to be more useful [77]. However, no clear favorite has emerged so far [78,79].

### 4.2. Antiinflammatory Therapy

IL-6 is the most widely studied inflammatory marker in SAH patients. The specifics of its involvement in the pathophysiology of the disease and its potential future use as a biomarker may be a matter of debate, however, all existing literature points to a central role for IL-6, and neuroinflammatory signaling in general, in the pathophysiology of SAH. This of course prompts speculations about the therapeutic potential of anti-inflammatory measures in the management of SAH patients.

Corticosteroids have been used for years in many centers in SAH patients. Overall, there is no clear evidence of beneficial (but also not of detrimental) effects [64,80,81,82]. An increased rate of medical complications was seen in the series reported by Miller et al. [83]. Of note, all relevant randomized studies are old and do not use dexamethasone. The more recent reports, more or less, all retrospectively compare outcomes and the incidence of DCI in patients receiving or not receiving variable doses of dexamethasone. The overall picture is quite similar to the discussion of the use of corticosteroids in neurotrauma. Their indiscriminate use in trauma patients has been proven detrimental [84], which possibly reflects corticosteroid-induced immunosuppression and infectious as well as other medical complications. On the other hand, this does not exclude the possibility that corticosteroids are helpful in selected cases, e.g., for the treatment of perilesional edema and increased intracranial pressure. The fact that very recent publications continue to investigate a potential role for corticosteroids in the management of SAH patients may indicate an ongoing interest, and that this issue has not been finally resolved.

A number of non-steroidal anti-inflammatory and antiplatelet drugs have also been investigated in SAH patients. This includes aspirin, ticlopidine, dipyridamole, and more recently, clopidogrel and COX2 inhibitors [82]. Of note, these investigations have often been motivated by the antithrombotic and antiplatelet rather than the anti-inflammatory effects of many of these drugs. Micro-thrombosis and platelet activation have been associated with DCI in SAH patients [10,11]. On the other hand, iatrogenic blood thinning and coagulation impairment in SAH patients carries substantial risks. These cases often require emergency surgical interventions, and ischemic brain and infarctions are at risk for hemorrhagic transformation. A 2007 Cochrane review of 1385 patients treated with various antiplatelet paradigms in seven randomized trials found no influence of these treatments on DCI and patient’s outcome [85]. However, there are some, albeit, retrospective studies reporting relatively large patient cohorts, which describe an association between the use of aspirin and clopidogrel and a reduced incidence of DCI [86,87,88]. This is good news since many neuro-interventional treatments for aneurysm obliteration require treatment with antiplatelet drugs. It might turn out that their use in SAH patients is not only relatively safe with respect to bleeding complications, but even beneficial.

A recent review by de Oliveira Manoel and Macdonald lists several other anti-inflammatory treatment paradigms investigated in the context of SAH [82]. This includes therapies that only secondarily target neuroinflammation. No conclusive benefits have been described. Outright immunosuppressive treatment with cyclosporine A has also been studied in a small series with SAH patients [89,90]. This concept has not attracted much interest, likely because of fear of an increased rate of infections in a patient population particularly prone to incur septic complications anyway.

Of note, all approaches mentioned are somewhat indiscriminate and target various aspects of neuroinflammation that may not necessarily involve IL-6. However, elevated IL-6 CSF levels following SAH is the common denominator of most studies investigating the role of neuroinflammation in SAH. Hence, it may be worth developing more IL-6-centered anti-inflammatory treatment paradigms. Intriguingly, the naturally occurring selective antagonist of IL-1 receptor antagonist IL-1Ra blocks upregulation of IL-6 and has been shown to protect against experimentally induced ischemia, traumatic brain injury and perinatal hypoxia in rodent models. In a proof of principle study involving 13 patients with acute SAH, Singh et al. showed that IL-1Ra indeed lowers IL-6 CSF levels [91]. A randomized trial with 136 participants replicated these findings using subcutaneous application of IL-1Ra [92]. The ongoing SC IL-1Ra in SAH trial (SCIL) is investigating subcutaneous application of recombinant IL-1Ra (Anakinrna, Kineret^®^) in a large prospective randomized multicenter series of SAH patients (ClinicalTrials.gov Identifier: NCT03249207). Results are not to be expected before 2023.

In summary, quite a few studies have already (sometimes unknowingly) attempted to use anti-neuroinflammatory therapeutic approaches in SAH patients in order to prevent or ameliorate ischemic brain injury, and specifically DCI. So far, no clear benefit has emerged. While this is somewhat disappointing, one can still hope that the so far negative data might reflect the mixed effects of unspecific immunosuppression or interfering with the wrong molecular pathway. An optimistic view would point to the many recent advances with targeted therapy approaches in, e.g., oncology and conclude that a better understanding of the molecular specifics of neuroinflammation in SAH will ultimately also result in successful treatment concepts.

### 4.3. Preclinical Studies

Many researchers have conducted preclinical investigations of IL-6 mediated neuroinflammation following SAH, some of which could have some translational potential. Several groups have focused on the characterization of the cellular component of the inflammatory response to SAH. As expected, resident microglia, but also monocyte and astrocyte activation and neuronal apoptosis paralleling increased IL-6 expression has been described in mice [93]. Inhibition of mast cell degranulation reduced IL-6 levels and relieved post-SAH injury in a rat model [94]. Suppression of neuroinflammation and IL-6 levels counteracts cerebrovascular endothelial dysfunction following SAH [95,96]. Investigating cellular rather than molecular targets for interventions in SAH patients is an innovative idea that may help to delineate new therapeutic concepts. The IL-6 mediated inflammatory response to SAH is apparently compartmentalized (CNS vs. systemic). It may prove useful to take this concept further down to the cellular level. One is also tempted to speculate about cell-based therapies [97].

Other researchers have attempted to identify and characterize the relevant upstream pathways resulting in (IL-6)-mediated neuroinflammation. Several studies describe pathways converging on raf-1/MAPK signaling [98,99,100], and ultimately, NF-κB [101,102,103,104]. Other signal transduction pathways have also been implicated [105,106]. Delineating signaling pathways and their targets is promising since specific inhibitors, and sometimes even drugs approved for use for other indications, are often already available [102] and can be potentially subjected to clinical trials.

Finally, some promising preclinical investigations have simply followed the concepts already explored in clinical studies. Wang and co-workers have reported positive effects in mice for the immunosuppressant fingolimod, which is commonly used in patients with multiple sclerosis [107]. The phosphodiesterase 4 inhibitor, roflumilast has anti-inflammatory properties and is prescribed for chronic obstructive pulmonary disease. Its use in a rat SAH model resulted in better neurological outcomes and lower levels of cerebral inflammation (including IL-6 expression) [108]. Most recently, Croci et al. have followed the paradigm of selective IL-6 inhibition [83,84] and describe encouraging results in a rabbit SAH model using the IL-6 antagonist tocilizumab [109].

## 5. Conclusions

The pleiotropic role of neuroinflammation in general, and IL-6 signaling in particular, renders IL-6 a very interesting marker for the monitoring of various conditions in SAH patients and also a potential target for therapeutic interventions. Indeed, there already seems to be a relatively large body of data linking elevated IL-6 CSF levels with at least some aspects of DCI and patient outcomes. However, translation of these findings into robust clinical applications is still a work in progress. Studies investigating IL-6 as a potential biomarker suffer from issues related to patient selection, the timing of IL-6 testing and also the definition of proper clinical endpoints such as DCI. Intervention studies have mainly used unspecific anti-inflammatory or non-IL-6-centered paradigms. We hope that future research will successfully address these issues. In addition, there is the promise that a better understanding of the molecular basis behind neuroinflammation in general, and the role of IL-6 signaling in particular, will ultimately result in clinical progress, since there can be no doubt whatsoever that neuroinflammation plays a central role in SAH pathophysiology.

## Figures and Tables

**Table 1 ijms-22-04133-t001:** Studies investigating IL-6 as biomarker for delayed cerebral ischemia (DCI) and functional outcomes after aneurysmal subarachnoid hemorrhage (SAH).

Study	Number of Patients ^1^	Timing ^2^	Marker	DCI ^2,3^	Outcome ^2,3,4^
Gaetani et al., 1998 [35]	31 SAH 10 controls	d0; surgery	CSF IL-6	+ (if surgery <72 h)	ND
			CSF IL-8, MCP-1, E-selectin	−	ND
Gruber et al., 2000 [36]	44 SAH	d0–2, 3–5, 6–8, 9–11, 12–14	CSF & serum IL-6	−	-
			CSF & serum sTNFR-1, TNF-α, IL-1β	−	-
Fassbender et al., 2001 [37]	35 SAH 20 controls	d1, 2, 3, 5, 7, 9, 11	CSF & serum IL-6	- (+ for CSF IL-6: TCD findings)	-
			CSF & serum TNF-α, IL-1β	−	-
Kwon et al., 2001 [38]	19 SAH 12 controls	d0; admission	CSF IL6	+	ND
			CSF TNF-α, IL-1β	+ (CSF IL-1 β)	ND
Schoch et al., 2007 [39]	64 SAH	daily	CSF IL-6	+ (d4-5)	ND
Nakahara et al., 2009 [40]	39 SAH	d3, 7, 14	CSF IL-6	ND	+
			CSF IL-8, HMGB-1, TNF-α	ND	+ (all markers)
Sarrafzadeh et al., 2010 [41]	38 SAH	d0, 1 (3x), 2–10 (2x daily); admission	CSF, serum & micro-dialysis IL-6	+ (CSF & micro-dialysis IL-6)	-
Muroi et al., 2011 [42]	99 SAH 20 BPH	daily	serum IL-6	+ (statistical trend)	ND
Ni et al., 2011 [43]	46 SAH	daily	CSF IL-6	+ (d3)	-
Chou et al., 2012 [44]	52 SAH	d0–1, 2–3, 4–5, 6–8, 10–14	serum IL-6	− (+ for angiographic vasospasm)	-
			serum TNF-α	− (+ for angiographic vasospasm)	+
McMahon et al., 2013 [45]	149 SAH	every other day	serum IL-6	+	ND
			WBC, ESR	+	ND
Muroi et al., 2013 [46]	138 SAH	daily	serum IL-6	+ (d3–7)	+ (d3–7)
			WBC, serum PCT & CRP	−	-
Höllig et al., 2015a [47]	81 SAH (46 with CSF analyses)	serum: d0, CSF: d1; admission	CFS & serum IL-6	ND	+ (CSF & serum)
			CSF & serum LIF, E-selectin, ICAM-1, MMP-9, serum CRP & WBC	ND	+ (CSF & serum LIF)
Höllig et al., 2015b [48]	53 SAH	d0, 1, 4, 7, 10, 14	serum IL-6	ND	+
			serum DHEAS	ND	+
Kao et al., 2015 [49]	53 SAH	d0; aneurysm coiling	serum (during coiling: aneurysm & peripheral. venous) IL-6	ND	+
Tang et al., 2015 [50]	58 SAH	d1, 4, 9	serum IL-6	+ (for DCI & angiographic vasospasm)	+
			serum ADAMTS13, vWF, P-selectin	+ (ADAMTS13: inverse correlation)	+ (ADAMTS13: inverse correlation)
Wu et al., 2016 [51]	57 SAH 65 controls	d2; admission	CSF IL-6	+	ND
			CSF TNF-α	+	ND
Chamling et al., 2017 [52]	89 SAH	d0, 1, 4, 7, 10, 14	serum IL-6	+ (d0 serum IL-6: inverse correlation)	ND
			serum LIF, E-selectin, ICAM-1, MMP-9, CRP & WBC	+ (WBC, CRP, LIF: inverse correlation)	ND
Chaudhry et al., 2017 [53]	80 SAH	d1, 3, 5, 7, 9, 11, 13	serum IL-6	+ (for DCI & angiographic vasospasm)	+
Kiiski et al., 2017 [54]	47 SAH	0, 12, 24 h, d1, 2, 3, 4, 5	serum IL-6	ND	-
			serum HMGB-1	ND	-
Lenski et al., 2017 [55]	63 SAH	daily	CSF & serum IL-6	+	ND
			serum CRP, & WBC CSF PMN% (% polymorphonuclear cells)	−	ND
Zhong et al., 2017 [31]	89 SAH 12 controls (surgery for incidental aneurysms)	d0; admission	serum IL-6	+	+
			serum IL-1 β, IL-2, IL-8, IL-10, CRP, T cell subpopulations	−	+ (IL-10)
Ďuriš et al., 2018 [32]	47 SAH	d1–4 (3x daily)	CSF & serum IL-6	−	+
			CSF & serum TNF-α, IL-1β	−	-
Wang et al., 2018 [56]	43 SAH 23 controls	d1, 4, 7, 10	CSF & serum IL-6	+	ND
			CSF & serum CRP, S100, NO, MMP-9	+	ND
Vlachogiannis et al., 2018 [57]	44 SAH	d1, 4, 10	CSF & serum IL-6	−	-
Ahn et al., 2019 [58]	60 SAH	d0, 1–2, 3–5, 6–8	serum IL-6	−	+ (d1–2)
			serum, 40 cytokines	+ (d0 IP-10, d1–2 PDGF-ABBB, d1–2 & d3–5 CCL5, d6–8 MIP1α)	+ (d6–8 MCP-1)
Al-Tamimi et al., 2019 [59]	43 SAH (29 with CSF analyses)	d1–3, 5, 7, 9	CSF & serum IL-6	−	+ (d3 serum IL-6, only univariate)
			CSF & serum IL-1α, IL-1β, IL-4, IL-8, IL-10, IL-15, IL-17, IL-18, MCP-1, VEGF, TNF-α	+ (CSF IL-4)	-
Rasmussen et al., 2019 [60]	90 SAH	d3, 8	serum IL-6	-	-
			IL-8, IL-10, ICAM-1, VCAM-1, IFNγ, TNF-α, hs (high sensitivity) CRP	− (+ for hsCRP: angiographic vasospasm)	-
Bjerkne Wenneberg et al., 2021 [61]	64 SAH	d0, 10; admission	serum IL-6	−	+ (d0)
			serum IL-1Ra, TNF-α, ICAM-1, CRP, WBC, PLT	−	+ (d0 serum IL-1Ra, TNF-α, CRP; d0 serum: ICAM-1, WBC, PLT, only univariate; d10 CRP)
Ridwan et al., 2021 [33]	82 SAH	d4–14 (≥3x), overall ≥6x	CSF & serum IL-6	− (+ for CSF IL-6: DCI with secondary infarct)	+

^1^ BPH, benign perimesencephalic hemorrhage; CSF, cerebrospinal fluid. ^2^ d0, 1, 2 etc., day 0, 1, 2 etc. following SAH unless otherwise specified (e.g., during surgery or neuroradiological intervention, at the time of hospital admission irrespective of the day of the hemorrhage). ^3^ +, −, ND; correlation between elevated marker and DCI/outcome, no correlation, not done; (..); further details regarding timing, marker and/or endpoint. ^4^ outcome; functional outcome (mRS = modified Rankin Scale or GOS = Glasgow Outcome Score).

## Data Availability

Not applicable.

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
