# Peer review of "Interleukin 6 and Aneurysmal Subarachnoid Hemorrhage. A Narrative Review"

_ijms, 2021, doi:10.3390/ijms22084133_

Round 1

Reviewer 1 Report

This is an interesting and timely review about the role of IL-6 in the multifaced disease subarachnoid hemorrhage. I suggest that the authors clearly state that this is a narrative review (in the title). Otherwise, if this review was performed systematically, they should follow the PRISMA guidelines and provide search algorithm (flowchart). Furthermore, I suggest (at least in the discussion section) that they add a Paragraph discussing preclinical work that provide further evidence to support their conclusion.

Author Response

"This is an interesting and timely review about the role of IL-6 in the multifaced disease subarachnoid hemorrhage. I suggest that the authors clearly state that this is a narrative review (in the title)..."

We indeed performed a narrative review. We have changed the title of the manuscript accordingly as suggested by the reviewer (lines 2-3).

"... Furthermore, I suggest (at least in the discussion section) that they add a Paragraph discussing preclinical work that provide further evidence to support their conclusion."

We agree with the reviewer. Therefore, we have changed the title of section 4 from "Clinical perspectives" to "Perspectives" (line 240) and included a new paragraph entitled "4.3. Preclinical studies" in the revised manuscript (lines 386-415). The new paragraph provides an account of the preclinical studies performed in the field which may have some translational relevance. This has resulted in the inclusion of 17 novel references in the manuscript (lines 692-742). We hope that this also addresses the reviewer's concern hat we may not have provided adequate and adequate references ("Are there appropriate and adequate references to related and previous work? 2 of 5").

Reviewer 2 Report

This is a complete and good review of IL-6 in SAH patients. The review describes in a very well organized structure the different possible roles of this molecule in these complex patients. I therefore suggest it should be accepted as it is. 

Author Response

Reviewer 2 had no concerns. We would like to thank the reviewer for her or his time.